# META ATTENTION FOR OFF-POLICY ACTOR-CRITIC

## ABSTRACT

Off-Policy Actor-Critic methods can effectively exploit past experiences and thus they have achieved great success in various reinforcement learning tasks. In many image-based and multi-source tasks, attention mechanism has been employed in Actor-Critic methods to improve their sampling efficiency. In this paper, we propose a meta attention method for state-based reinforcement learning tasks, which combines attention mechanism and meta-learning based on the Off-Policy Actor-Critic framework. Unlike previous attention-based work, our meta attention method introduces attention in the actor and the critic of the typical Actor-Critic framework rather than in multiple pixels of an image or multiple information sources. In contrast to existing meta-learning methods, the proposed meta-attention approach is able to function in both the gradient-based training phase and the agent's decision-making process. The experimental results demonstrate the superiority of our meta-attention method in various continuous control tasks, which are based on the Off-Policy Actor-Critic methods including DDPG, TD3, and SAC.

## 1 INTRODUCTION

Reinforcement Learning (RL) algorithms based on the Actor-Critic framework have achieved considerable success in many areas such as games, robot control, and planning. Compared with on-policy methods, off-policy methods possess more efficient sampling since they do not require new samples to be collected for each gradient step and make better use of experience (Haarnoja et al., 2018a). However, even for off-policy methods, traditional reinforcement learning algorithms still have extremely low sample efficiency (Yu, 2018). Recently, meta-learning (Hospedales et al., 2020) has become topical as a paradigm to accelerate RL by learning an inductive bias from past experience. By learning aspects of the learning strategy, such as fast adaptation strategies (Finn et al., 2017) (Rakelly et al., 2019), losses (Zhou et al., 2020) (Bechtle et al., 2020), optimization strategies (Duan et al., 2016), exploration strategies (Gupta et al., 2018), hyperparameters (Bechtle et al., 2020), and intrinsic rewards (Zheng et al., 2018), meta-learning has significantly improved sample efficiency over standard RL algorithms.

Improving sample efficiency through attention mechanism has also been proved to be very effective in image-based reinforcement learning(Barati & Chen, 2019; Chen et al., 2019). The application of attention mechanism in multi-agent system(Parnika et al., 2021; Iqbal & Sha, 2019) and multi-object task(Team et al., 2021) also shows its powerful capabilities in information processing. However, in the existing works, attention mechanisms often have clear application scenarios, such as image-based control or multiple sources of information (multi-agent or multi-target). The effective combination of attention mechanisms with current algorithms in a state-based single-agent environment is still a problem to be investigated

In this paper, we propose a meta attention method based on the attention mechanism. In the human decision-making process, people often modify their concepts based on the feedback and results to obtain a better decision. Inspired by this decision-making process, we use meta attention to adapt the features generated by the policy network based on the evaluation of value network. Our work differs from current attention-based work in that our meta attention approach works only within the Actor-Critic framework and does not depend on a specific scenario. We formalize the meta-learning process as a bi-level optimization problem. Our approach can be flexibly combined with various algorithms by using meta attention as a meta-learner and optimizing meta attention in the outer layer. Unlike the existing meta-learning methods, our meta attention approach can improve the

performance of agent through gradients in the training stage and obtain better actions by adjusting the features in the execution stage.

We evaluated the proposed meta attention method in a series of continuous control tasks in Gym and Roboschool, including three 3D robot control tasks, two 2D control tasks, and one classic control task based on DDPG(Lillicrap et al., 2016), TD3(Fujimoto et al., 2018) and SAC(Haarnoja et al., 2018b). Besides, we also discussed the changes and impact caused by modifying the actor features through meta attention. Experimental results how that our meta-attention approach is not only effective in accelerating the learning progress of the agent in the training phase, but also improves the actions in the execution phase, further enhancing the performance of the agent.

## 2  RELATED WORK

**Attention Mechanism**  Attention is a behavioral and cognitive process of selectively attending to a discrete aspect of information, whether subjective or objective, while ignoring other perceptible information(de Santana Correia & Colombini, 2021). Typically, attention mechanism is mainly applied in the fields of computational vision and natural language processing. Accordingly, although the implementation methods are different, the application of attention mechanism in reinforcement learning is mainly focused on video games(Wu et al., 2021; Chen et al., 2019; Barati & Chen, 2019; Mott et al., 2019)(Manchin et al., 2019). Other works such as Peng et al. (2020) proposed a dynamic attention model with a dynamic encoder-decoder architecture, which dynamically explores node features to efficiently exploit hidden structural information at different construction steps. Li et al. (2021) applied the attention mechanism to generate feature vectors and input them into value and policy head during the feature extraction phase of PPO and PPG.Jiang & Lu (2018); Iqbal & Sha (2019); Mao et al. (2019) employed a multi-head attention mechanism to make one agent selectively pay attention to information from other agents. Team et al. (2021) used the attention mechanism to match multiple goals with the hidden state of the current state to obtain the goal-attention hidden state under different goals. This approach allows the agent to predict the expected return obtained by attending to a goal at the end of a scene.

**Meta Reinforcement Learning**  Meta-learning is most often understood as learning to learn, which refers to learning from historical information or multiple learning episodes to improve learning algorithms. Since works which fed historical trajectories into Recurrent Neural Network to flesh out task-level information(Wang et al., 2016; Duan et al., 2016), various meta-learning methods have been proposed to strengthen agent performance. Houthooft et al. (2018); Kirsch et al. (2020); Zhou et al. (2020) used meta-learning methods to learn a loss function rather than artificial design to improve performance of agent in single or multiple tasks.Gupta et al. (2018); Stadie et al. (2018); Xu et al. (2018a) employed meta-learning methods to learn exploration instead of traditional exploration methods.Finn et al. (2017) meta learned a good model initialization of the model that can be quickly adapted to different tasks. Xu et al. (2018b) improved the performance of the agent by meta-learning discount factors.Rakelly et al. (2019)Fakoor et al. (2020) treats meta information as an unobservable state of Partially Observable Markov Decision Process, further improves the agent's performance in multi task learning. Although the attention mechanism may not have an explicit meta-learning object, it can also be considered as the king of the meta-learning method.

**Bi-level Optimization**  Generally, meta-learning can be formalized as a bi-level optimization(BLO) problem. However, the solution of BLO problems is often challenging. Franceschi et al. (2018) proposed a framework for approximating the solution of BLO problems using the gradient method. Since then, many works based on the gradient methods to optimize meta leaner have successfully proved the feasibility of gradient method, such asLi et al. (2019); Finn & Levine (2018); Lian et al. (2020); Flennerhag et al. (2020). For the RL problem, Zhou et al. (2020) optimize the meta critic, the upper-level of a BLO, as an intrinsic motivation by gradient method. Kirsch et al. (2020) enables a population of agents to use and improve a single parameterized objective function through gradient learning on different tasks. Rajeswaran et al. (2019) proposed an implicit MAML algorithm by drawing upon implicit differentiation that effectively decouples the meta-gradient computation from the selection of an inner loop optimizer. Liu et al. (2019) proposed a surrogate objective function TMAML, which incorporates control variables into gradient estimation through automatic

differentiation and improves the quality of the gradient estimation by reducing the variance without introducing bias.

## 3 METHODOLOGY

### 3.1 OFF-POLICY ACTOR-CRITIC

In general, the reinforcement learning task can be considered as finding the optimal policies in Markov Decision Processes (MDPs). The MDP is defined by a tuple $(\mathbb{S}, \mathbb{A}, \mathbb{P}, \mathbb{R})$, where $\mathbb{S}$ is a set of states, $\mathbb{A}$ is a set of actions, $\mathbb{P}$ is a set of probabilities to switch from a state $s$ to $s'$ for a given set of action $a$, and $R : \mathbb{S} \times \mathbb{A} \to \mathbb{R}$ is a scalar reward function. In the Actor-Critic framework, the policy network(Actor) $\pi_\phi(s)$ and the value network(Critic) $Q_\theta(s, a)$ are parameterized by a neural network respectively. At each time $t$, the agent receives an observation $s_t$ and takes a action $a_t$ based on its policy $\pi : \mathbb{S} \to \mathbb{A}$, then receives a reward $r_t$ and a new state $s_{t+1}$. The tuple $(s_t, a_t, r_t, s_{t+1})$ describes a state transition and will be stored in a reply buffer $\mathcal{D}$ for off-policy learning. The objective of RL is to find the optimal policy $\pi_\phi$ to maximizes the expected cumulative return $J$:

$$J(\phi) = \mathbb{E}\left[\sum_{t=0}^{\infty} \gamma^t R\left(s_t, a_t\right) \mid a_t \sim \pi_\phi\left(\cdot \mid s_t\right)\right] \tag{1}$$

Where $\gamma$ is the discount factor, and $J(\phi)$ also can be written as the expected value for the Q-function:

$$J(\phi) = \mathbb{E}_{s \sim p_\pi} Q_\theta(s, a)|_{a_t \sim \pi_\phi(\cdot|s_t)}, \tag{2}$$

Where $Q$-function is the expected discounted sum of rewards following visitation at state s and execution of action $a$, and $p_\pi$ is the state distribution induced by policy $\pi$. For off-policy Actor-Critic architectures such as DDPG, TD3 and SAC, the loss for the actor provided by the critic may be different, but the $Q$-function will be learned by minimizing the loss of the below equation as same:

$$L_\theta^{Critic} = \mathbb{E}_{s_t \sim p_\pi, a_t \sim \pi_\phi}\left[\left(Q_\theta\left(s_t, a_t\right) - y_t\right)^2\right]$$
$$y_t = r_t + \gamma Q_{\theta'}\left(s_{t+1}, \pi_{\phi'}\left(s_{t+1}\right)\right), \tag{3}$$

where $\phi'$ and $\theta'$ represent the target network for the critic and the actor respectively. The actor loss usually differs in details according to different algorithms, however , they all follow a form of the following formula:

$$L_\phi^{Actor} = -J(\phi) = -\mathbb{E}_{s \sim p_\pi} Q_\theta(s, a)|_{a = \pi_\phi(s)} \tag{4}$$

### 3.2 META ATTENTION METHOD

The attention mechanism has been widely used in image recognition and natural language processing since it can model the human pattern recognition process. It allocates attention to the important part of information while automatically ignores low-value features. According to (Vaswani et al., 2017), attention function can be described as mapping a query and a set of key-value pairs to an output, where the output is computed as a weighted sum of the values, with the weight assigned to each value being computed by a compatibility function of the query with the corresponding key:

$$\text{Attention(Query, Key, Value)} = \sum_i \text{Similarity}\left(Query_i, Key_i\right) * \text{Value}_i$$

In order to introduce the attention mechanism into the Actor-Critic framework, we first split the actor and critic network into two parts . We take the last layer of actor network $\hat{\pi}$ as an action net to produce actions and the rest as actor feature net $\bar{\pi}(s)$ to extract features, with the entire policy network being denoted as $\pi_\phi(s) = \hat{\pi}(\bar{\pi}(s))$. Similarly, the value network can also be divided into a critic net $\hat{Q}$ and a critic feature net $\bar{Q}(s)$, and the whole value network can be expressed as $Q_\theta(s) = \hat{Q}(\bar{Q}(s, a))$. We use the feature net as encoder to obtain Query, Key and Value by the following formula:

$$\text{Querry} = \bar{\pi}(s) \quad \text{Key} = \bar{Q}(s, \pi_\phi(s)) \quad \text{Value} = \bar{\pi}(s)$$

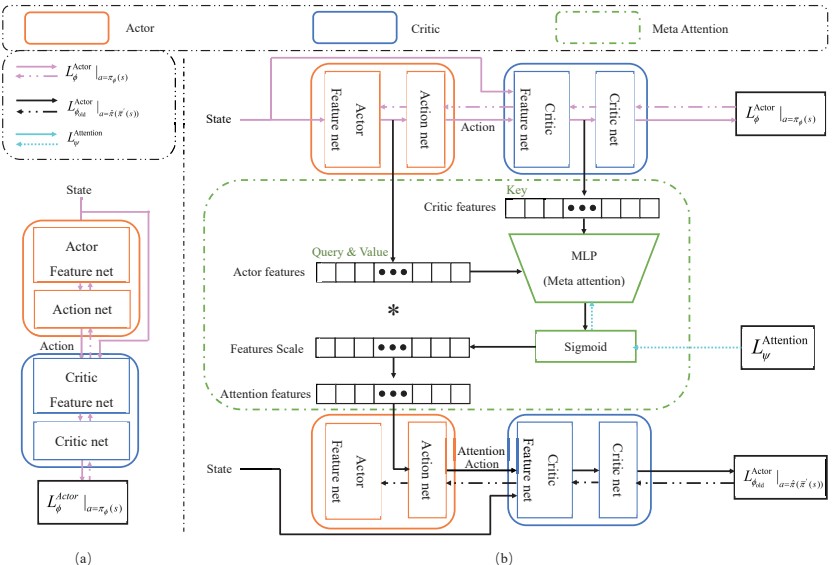

Figure 1: (a) Traditional Actor-Critic framework. (b) The meta attention Actor-Critic framework. The part circled by the orange box represents the actor, the part circled by the blue box represents the critic, and the part circled by a green dashed line represents the attention method. There is two orange(blue) boxes in (b), but they refer to the same actor(critic). Lines with different colors represent the update process dominated by different losses.

We input Query(actor features) and Key(critic features) into meta attention network $f_\psi(Query, Key)$ (a three-layers MLP parameterized by $\psi$) to calculate the similarity of each feature dimension. To enhance or reduce the features in specific corresponding dimensions, we multiply the output after the sigmoid function by 2 to obtain the feature scale. In this paper, we use $\circ$ to denote the Hadamard product, and by calculating the Hadamard product $\circ$ of Value(actor features) and scale, we obtain the attention features $\bar{\pi}'(s)$ after modifying some dimensions:

$$\text{Attention Features} = \bar{\pi}'_\psi(s) = f_\psi(\bar{\pi}(s), \bar{Q}(s, \pi_\phi(s))) \circ 2 \circ \bar{\pi}(s) \tag{5}$$

This calculation process corresponds to the part in the green dotted line box in Figure 1 (b).

There are also two critical problems in the optimization process. 1) how to affect the agent's decision-making and training process through attention features; and 2) how to optimize the meta attention network to achieve the correct matching Query(actor features) and Key(critic features) to generate the proper feature scales. To address these issues, we formalize the entire optimization process as a bi-level optimization problem, referring to the meta attention method as the outer level, and to the task performed by the agent as the inner level:

$$\psi^* = \arg\min_\psi L_\psi^{\text{Attention}}(\mathcal{D}; \phi^*; \psi) \tag{6}$$

$$\text{s.t. } \phi^* = \arg\min_\phi [L_\phi^{Actor}(\mathcal{D}; \phi | a \sim \pi_\phi(s)) + L_\phi^{Actor}(\mathcal{D}; \phi; \psi | a \sim \hat{\pi}(\bar{\pi}'_\psi(s)))], \tag{7}$$

where $L_\psi^{\text{Attention}}$ is the meta optimization objective, and $L_\phi^{Actor}$ is the actor loss in the form of Eq.(4).

For the first problem, we first implement a traditional back-propagate dominated by the actor loss $L_\phi^{\text{Actor}}|_{a=\pi_\phi(s)}$ on training data $d_{trn}$:

$$\phi_{old} = \phi - \eta \frac{\partial L_\phi^{\text{Actor}}(d_{trn} | a \sim \pi_\phi(s))}{\partial \phi}, \tag{8}$$

Where $\eta$ is the learning rate, this process corresponds to the upper part of Figure 1 (b) and the first step of meta-training in Algorithm1. Then, we generate a new attention action $a =$

$\hat{\pi}(\bar{\pi}'_\psi(s))$ from the attention features through action net and feed into the critic to get the loss $L_\phi^{\text{Actor}}\left(\mathcal{D};\phi;\psi\right)|_{a=\hat{\pi}(\bar{\pi}'_\psi(s))}$ for back-propagation:

$$\phi_{new} = \phi_{old} - \eta \frac{\partial L_{\phi_{old}}^{\text{Actor}}\left(d_{trn};\psi|a \sim \hat{\pi}(\bar{\pi}'_\psi(s))\right)}{\partial \phi} \tag{9}$$

This allows the agent to obtain better actions by simply modifying the features compared to the original action without increasing the batch size, strengthening the agent's learning of good actions and reducing the probability of producing poor actions after the gradient step. This process corresponds to the lower part of (b) in Figure 1 and the second step of meta-training in Algorithm1.

For the second problem, we have two basic assumptions: 1) a good meta attention network will inevitably generate feature scales that can generate actions with higher value since it better associates the relationship between actor features and critic features; 2) in the process of back-propagation, high-value actions enhance actors more than low-value actions because good actions will strengthen the actor's tendency to make such choices, while bad actions cause the actor to try other actions. Furthermore, since the meta attention network is involved in the back propagation process, we use the utility of meta attention in this process as the attention loss $L_\psi^{\text{Attention}}$ and the attention loss on validation data $d_{val}$ is defined as:

$$L_\psi^{\text{Attention}} = \tanh\left(L_{\phi_{new}}^{\text{Actor}}\left(d_{val}|a \sim \pi_{\phi_{new}}(s)\right) - L_{\phi_{old}}^{\text{Actor}}\left(d_{val}|a \sim \pi_{\phi_{old}}(s)\right)\right) \tag{10}$$

Under this definition, when performing gradient descent updates, it is ensured that the meta attention is always updated along the direction that improves the performance of the agent. This process corresponds to the blue line part of (b) in Figure 1 and the meta-test in Algorithm1.

### 3.3 BEYOND THE GRADIENT

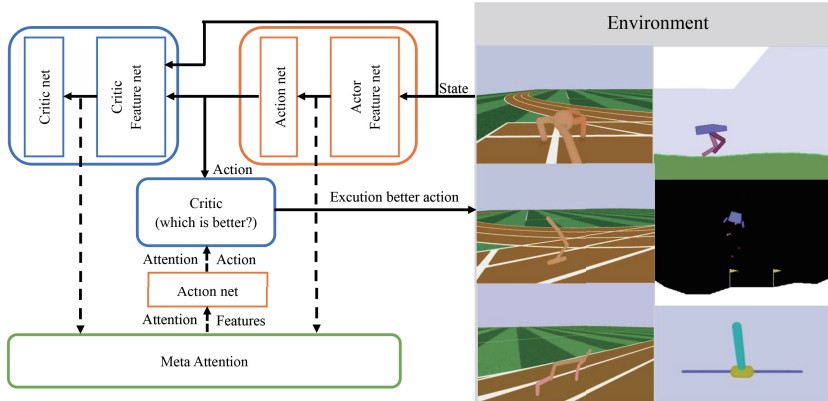

Figure 2: When the agent interacts with the environment, in addition to the action produced by the traditional Actor-critic framework, meta attention also attempts to make further adjustments to the initially produced results in the hope of achieving better results.

Unlike previous gradient-based meta-learning approaches, our proposed meta attention method can affect the learning process of the agent through the gradient and participate in each of their decisions. In the traditional Actor-Critic framework, the critic estimates each action's value based on the current state, which provides a criterion for evaluating the action produced by an actor. The Critic and the trained meta attention make it possible for meta attention methods to participate in decision-making.

When people are given the rules for scoring in human decision-making, they often modify their previous ideas to obtain better results. Similarly, we combine the attention method with the critic to introduce this decision-making method into the AC framework. For the deterministic policy, when an agent is given a state from the environment, we can get another attention action by adjusting the actor features in addition to the actions directly produced by the traditional Actor-Critic framework. Then the action with the higher Q value given by the critic is executed. This decision process is shown in Figure 2 and corresponds to the environment step in Algorithm 1.

This method of action selection is like an actor dancing once more after a performance, getting another score by adapting his or her actions based on the feedback given by the critics, and choosing which set of actions to be performed formally by comparing the two scores.

For stochastic policies such as SAC, their actions under a continuous control task are typically sampled from a Gaussian distribution. Although it is meaningless to compare sampled actions, we can still compare and modify the actor features in terms of the mean value of Gaussian distribution. Another attentional action is then sampled from the Gaussian distribution and the parameters (mean and variance) generated by the attentional features are modified.

Although the Q-value given by the critic may be inaccurate in the early stage of training, and off-policy algorithms typically have serious over estimation problem, these issues are beyond the scope of this paper.

---

**Algorithm 1** Meta attention for off-policy Actor-Critic

---

Initialized parameters $\theta, \phi, \psi$, $D = \emptyset$ and learning rate $\eta, \lambda$

  **for** each iteration **do**
    **for** each environment step **do**
      **if** the actor dances twice **then**
        $a_t \sim \pi_\phi\left(s_t\right)$ or $a_t \sim \hat{\pi}(\bar{\pi}'_\psi(s))$          %  Select the action with higher Q-value
      **else**
        $a_t \sim \pi_\phi\left(s_t\right)$          %  Traditional action generation process
      **end if**
      $s_{t+1} \sim p\left(s_{t+1} \mid s_t, a_t\right), r_t$       %  Observe reward $r_t$ and new state $s_{t+1}$
      $\mathcal{D} \leftarrow \mathcal{D} \cup \left\{(s_t, a_t, r_t, s_{t+1})\right\}$       %  Store the transition in the replay buffer
    **end for**
    **for** each gradient step **do**
      Sample mini-batch $d_{trn}$ from $\mathcal{D}$
      $L_\theta^{Critic} \leftarrow$ Eq.(3)
      $\theta \leftarrow \theta - \eta \nabla_\theta L_\theta^{Critic}$       % Critic update
      **meta-training:**
      $\phi_{old} \leftarrow$ Eq.(8)       % Vanilla Actor update
      $\phi_{new} \leftarrow$ Eq.(9)       % Actor update based on attention action
      **meta-test:**
      $L_\psi^{\text{Attention}} \leftarrow$ Eq.(10)       % Attention loss
      **meta-optimisation:**
      $\phi \leftarrow \phi_{new}$       % Update the actor parameters
      $\psi \leftarrow \psi - \lambda \nabla_\psi L_\psi^{\text{Attention}}$       % Update the meta attention parameters
    **end for**
  **end for**

---

## 4 EXPERIMENT

### 4.1 SETUP

To evaluate our meta attention framework, we selected four 3D continuous robot control tasks in Roboschool and two 2D continuous robot control tasks in Gym without modifying the original environments or rewards. Although MuJoCo is widely used to test various reinforcement learning algorithms, Roboschool provides a more realistic and difficult robot control task than MuJoCo. Compared with MuJoCo, Roboschool reduces the alive bonus and increases drive costs to encourage movement in a low-energy manner. Roboschool also made other adjustments, such as encouraging the Ant robot to have two or more legs on the ground to ensure that the robot moves in a more natural posture.

We applied our meta attention framework to DDPG, TD3 and SAC as the basic algorithms. Algorithms that use meta attention methods only in the training stage will be annotated as _**MATT** (such as DDPG_MATT), while algorithm that uses the meta attention method in both the training stage and agent decision-making process will be annotated as _**MATT_DT**(such as DDPG_MATT_DT).

We also compare our _**MATT** and _**MATT_DT** with another SOTA work (Zhou et al., 2020) which proposed Meta Critic (_**MC**) to enforce the basic algorithm.

All our codes are built on the code provided by Zhou et al. (2020). We ran tasks for 1 million or 5 hundred thousands game steps depending on the environment, and we evaluate our policy over 10 episodes without exploration every 1000 game steps. All results for each task are averaged over 10 random seeds (trials) and network initializations. 95% confidence intervals are shown as shaded regions over time steps. Following Fujimoto et al. (2018), curves are smoothed evenly for clarity (window_size=10).

Experiments in all environments had the same network structure and hyperparameters, with a learning rate of 0.003 for DDPG and TD3 but 0.001 for SAC. All experiments are performed on 4 servers with 4 NVIDIA-TITAN-V GPU each and 3 servers with 6 NVIDIA-TITAN-V GPUs. Each server is equipped by 2 Intel(R) Xeon(R) Gold 5218 CPUs.

## 4.2 EVALUATION OF META ATTENTION METHOD

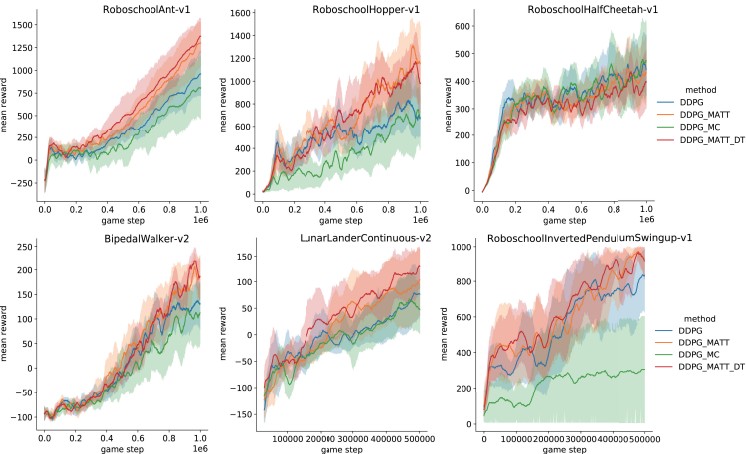

Figure 3: The learning curves of DDPG and enhancement algorithms. The shaded region represents the 95% confidence interval.

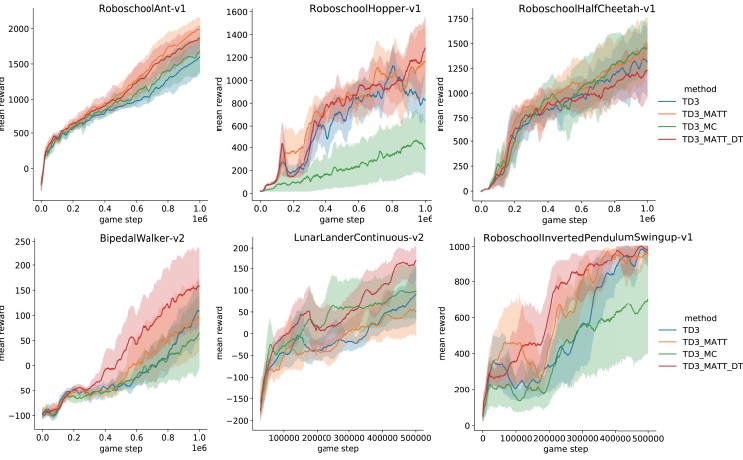

Figure 4: The learning curve of TD3 and enhancement algorithms. The shaded region represents the 95% confidence interval.

**Deterministic policy** Figure 3 and Figure 4 demonstrate the comparison between the vanilla algorithms and their enhancive versions based on DDPG and TD3. From the figures, we

observes that _MATT and _MATT_DT achieve significant performance improvements in all but the *RoboschoolHalfCheetah-v1* environment. **DDPG_MATT** perform almost as well as **DDPG_MATT_DT** under the environments *BipedalWalker-v2*, *LunarLanderContinuous-v2* and *RoboschoolInvertedPendulumSwingup-v1*. But in these enviroment, **TD3_MATT_DT** perform significantly better than **TD3_MATT**, and we speculate that this phenomenon may be related to more accurate Q-values.

Similar to Meta Critic, _MATT and _MC only work in the training stage and influence the actor through gradients, but **DDPG_MC** performs even worse than the original algorithm in four environments, while **DDPG_MATT** shows performance improvements in all environments except *RoboschoolHalfCheetah-v1*. **TD3_MATT** also shows high performance in more environment than **TD3_MC**. From the data provided in Zhou et al. (2020), we can also find a similar situation: the performance of _MC decreases in the first 1 million steps in deterministic policy based algorithms. This also shows that our meta attention method completes the meta-learning process faster than Meta Critic.

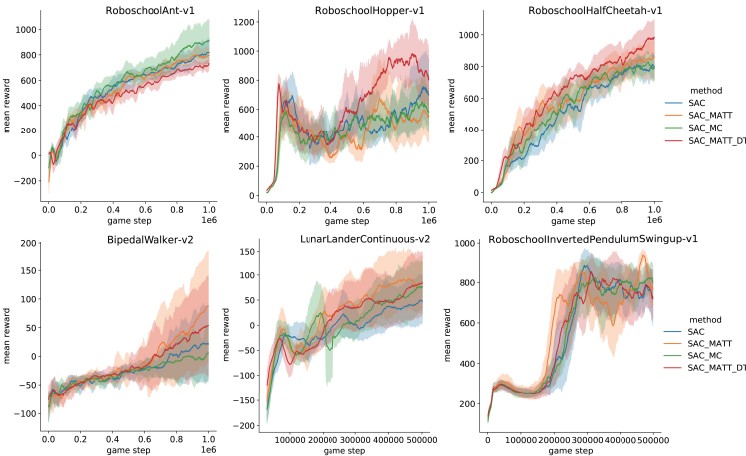

Figure 5: The learning curve of SAC and enhancement algorithms. The shaded region represents the 95% confidence interval.

**Stochastic policy** Figure 5 compares the SAC and its enhancive versions. Although **SAC_MATT** has only achieved performance improvement in environments *BipedalWalker-v2* and *LunarLanderContinuous-v2*, **SAC_MATT_DT** still achieves performance improvements in all environments other than *RoboschoolAnt-v1* and *RoboschoolInvertedPendulumSwingup-v1*. This suggests that our method is also effective in stochastic policy, but the improvement in stochastic policy is less pronounced than in Deterministic policy. We speculate that, compared with deterministic strategies, sampled actions in random strategies bring more uncertainty to the learning of meta attention. The performance of **SAC_MATT** and **SAC_MC** in the four environments is similar, which is the same as or slightly higher than the original algorithm. The performance of the first 1 million steps also matches the data provided in the Zhou et al. (2020).

## 4.3 FURTHER ANALYSIS

In order to analyze the effect of meta attention on the features, we ran 5 random seeds(trails) in *RoboschoolWalker2d-v1* environment. In each trial, we ran a total of 1 million game steps and evaluate our policy over 10 episodes without exploring every 1000 game steps. Figure 6 (a) shows an overview of this experiment, and the shaded region represents the 95% confidence interval.

We recorded the Q-values of attention and actor features during the evaluation stage to verify the impact of meta attention on performance. In the evaluation phase, we took 1 million consecutive Q-values of the actions generated by attention features and actor features. These 1 million samples correspond to 1 million time steps in the evaluation stage or 10 evaluations. We used the error between Q-values of the attention action and vanilla action as performance improvements brought

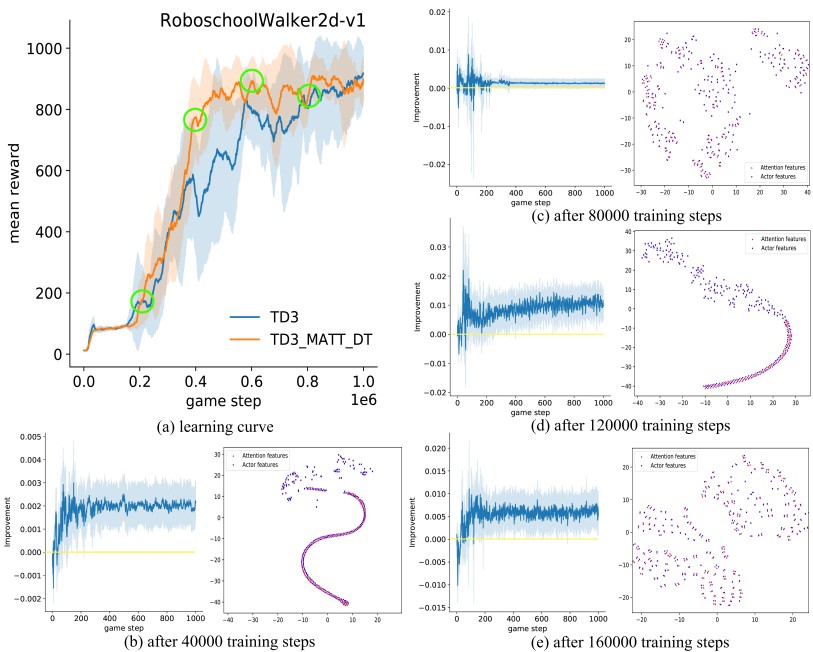

Figure 6: (a) Learning curves, the shaded part represents the 95% confidence interval; (b-e) The results obtained by sampling at different stages (green circle in (a)) of training. The left part represent the difference in Q-values of actions generated by attention features and actor features, respectively The right part represents t-SNE visualization of attention features and actor features, where red dots represent attention features and blue dots represent actor features.

by meta attention. We sampled at four different time points in the training phase, and these results are represented in Figure 6 (b-e).

To analyze the diversity of attention features and actor features, we sampled 250 attention features and actor features on continuous 250 time steps during evaluation on one trail. We used t-SNE to visualize these total 500 features. Similarly, we sampled at five different time points in the training stages, and these results are shown in the right part of Figure 6 (b-e).

Figure 6 (b-e) shows the results of sampling at approximately 40000, 80000, 120000, 160000 training steps on one trail of 1 million time steps or 200000 training steps. In terms of improvement(left part of Figure 6 (b-e)), the actions obtained after the modification of our meta attention usually achieved higher Q-values at different stages of training. In terms of the distribution (right part of Figure 6 (b-e)), attention features have changed in most cases compared with actor features, as shown by the apparent separation of red points and blue points. This suggests that our meta-attention approach is effective in modifying features and improving performance in the current state throughout the training phase.

## 5 CONCLUSION

In this paper, we present the meta attention method, a derivative-based meta learner for off-policy Actor-Critic reinforcement learning methods. By modifying the actor features, we speed up the training process and enable agents to make better decisions when interacting with the environment. The experimental results demonstrate the effectiveness of our method in both stochastic policy and deterministic policy. Moreover, our method can be flexibly integrated into a variety of contemporary off-policy Actor-Critic methods to boost performance. In future work, we will use the multi-head attention to improve the meta learner and decouple the meta attention method from the value function to reduce the impact of the over estimation problem.

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
