# OpenReview forum: "Meta Attention For Off-Policy Actor-Critic"
_ICLR.cc/2022/Conference — ICLR 2022 Submitted_

### Official Review · Reviewer_R4EG · 2021-10-27

**Correctness:** 2
**Technical Novelty And Significance:** 3
**Empirical Novelty And Significance:** 2
**Recommendation:** 5
**Confidence:** 3

**Main Review:**

Strength:
1. The paper brings an interesting perspective to RL. The attention mechanism combines the information from the actor and the critic to further refine the actor representation. This is also supported by better empirical performance from the proposed method.
2. The paper is mostly clearly written and easy to follow, with informative diagrams for explaining the proposed method.

Weakness:
1. The proposed algorithm does not exactly match the bi-level optimization formulation from Eq. 6 and 7. In particular, Eq. 6 is broken down into two separate gradient steps. In addition, argmin from Eq. 7 is approximated by a single gradient step. Lastly, \phi^* is not used during meta training. Instead, the algorithm requires \phi_old and \phi_new as input for meta training.
The issues above makes it challenging to understand the proposed method from a conceptual level: what exactly is meta attention optimizing for?

2. I find the several design choices ad-hoc and not well motivated. Specifically,

      a. The attention scaling factor of 2 is not explained.

      b. The use of tanh in the meta-learning loss is not a standard choice and deserves more justification.

      c. The meta-learning loss considers actions produced by \phi_new and \phi_old, but not the action produced by the attention mechanism. Could the authors further elaborate on this choice?

3. The presentation for the empirical results is too dense (figures only), and without any numerical comparison. It is difficult to assess how much better than proposed method is compared to the baselines. It's also difficult to assess the stability of different methods, when the figures appear to show significant standard deviation of performance for the meta-trained methods. Could the authors provide numerical comparison of the experiments?

4. It is difficult to interpret the results between _MATT and _MATT_DT variants. _MATT seems to outperform in some settings while under-perform in the rest. Most crucially, Neither _MATT nor MATT_DT exactly matches what the model is trained on (e,g. there is an extra selection step for _DT). Could the authors elaborate on the comparison between the two variants? Could we compare with the attention action without the extra selection step?

5. There is no discussion on the additional computational complexity imposed by the meta-learning formulation. It would be good to compare the computational resources needed, as well as the sample complexity of different methods.

**Summary Of The Paper:**

The paper introduces attention mechanism into actor-critics method, and formulates RL as a bi-level optimization to learn the (meta) attention parameters. The attention mechanism appears model agnostic and acts on the feature representation from the actor and critic models. Empirically, the proposed model shows improved performance over baseline methods.

**Summary Of The Review:**

The proposed method brings an interesting perspective combining RL and meta-learning, and demonstrates better empirical results. However, the model design is not well motivated, and the optimization objective does not match the practical algorithm. Lastly, the empirical results are too concise for detailed performance assessment. I think the current paper would benefit from another revision.

---

### Official Review · Reviewer_v8k3 · 2021-11-01

**Correctness:** 3
**Technical Novelty And Significance:** 3
**Empirical Novelty And Significance:** 2
**Recommendation:** 5
**Confidence:** 4

**Main Review:**

Strengths:
The paper is generally well-written. Figure 1 and Figure 2 clearly present the main idea of this work and it helps understand this work.
It is relatively novel to consider the idea of applying attention mechanism between actor and critic instead of only for image feature extraction.
The proposed method can be easily used in any off-policy actor-critic network and the experiments show the advantage of the proposed method on continuous control tasks.

Weaknesses:

It seems that this work positions itself in the literature of meta-learning. They call it "meta-attention" (in abstract), "meta learner" (in conclusion). Considering that most works in meta-learning investigate the multi-task setting. It might be better to add experiments of the proposed method on meta RL benchmarks? Please correct me if this impression or understanding of the proposed method is wrong.

It is unclear to me, what does the proposed attention mechanism learn to pay attention to? Consider the image-based attention mechanism or the multi-object attention mechanism, the attention is used to extract the important region of an image or an important object in the set of objects. In this work, the attention weight is calculated for each feature dimension. Then why should some dimensions be more important than the other dimensions? What information is represented in each dimension? Could you please provide an intuitive explanation about the meaning of attention in the proposed method?

Another question is also related to the attention mechanism. It is said: "meta attention network to calculate similarity of each feature dimension". Why a three-layer MLP could calculate the similarity of actor feature and critic feature? How is the similarity defined?

In equation 10, why we should use tanh instead of other possible functions to formulate the loss for attention? Is there any intuition behind it?  Did you try other design choices of attention loss?

It is said the challenges of value estimation in RL are "beyond the scope of this paper". This point is not very convincing to me. The proposed method employ critic features to adapt the actor features (through the attention mechaism) and expect that the new action features are better for actor neural network. This means the advantage of the proposed method heavily relies on good critic estimation. When the critic estimation is problematic or not accurate enough, will the proposed attention mechanism even harm the performance of the standard off-policy actor-critic? This point will definitely limit the application of the proposed method to more challenging RL problems. I think this might be an important weakness of the proposed idea.

**Summary Of The Paper:**

This paper proposes to modify the off-policy actor-critic framework by introducing an attention mechanism in the actor and critic. The attention mechanism is used to adjust the actor features (i.e. intermediate features generated by the actor neural network) for better action selection in the continuous control tasks. The query is the actor feature. The key is the critic feature. The output from the attention network is the similarity of each feature dimension. The new actor feature is the attention-weighted action feature and it is used to generate new action in the actor neural network.

Experiments demonstrated that the proposed attention mechanism between actor and critic network can improve actor-critic algorithms, DDPG, TD3, and SAC.

**Summary Of The Review:**

The intuition and meaning of the attention mechanism need more explanations. The advantage of the proposed method seems to rely on an accurate critic network and this advantage may become a disadvantage when the critic is not good enough.

---

### Official Review · Reviewer_ihtm · 2021-11-02

**Correctness:** 2
**Technical Novelty And Significance:** 2
**Empirical Novelty And Significance:** 1
**Recommendation:** 3
**Confidence:** 4

**Main Review:**

The basic idea of this paper is quite interesting; it introduces an attention mechanism that tries to exploit differences in representations between the policy and the value function to improve the policy. The attention mechanism is also learned in an interesting fashion, as it is formulated as a meta-learning problem: the goal of the attention layer is to be such that it improves learning of other components of the agent. Empirically, the authors demonstrate that the algorithm does yield performance gains.

With that said, this paper is not ready for publication. A minor issue is that the writing makes several sweeping statements that cannot be verified or that have no clear meaning. As an example, the authors motivate their work in the abstract by the phrase "our meta attention method introduces attention in the actor and the critic of the typical Actor Critic framework rather than in multiple pixels of an image or multiple information sources". This is clearly false, as the most common way to use attention is either in the form of a transformer or as neural memory, neither of which operate directly on pixels.

Similarly the technical presentation is imprecise, which both calls in the question what the authors are actually proposing and whether the exposition is correct. Concretely, the presentation of the RL framework is not complete and somewhat incorrect; the true return in Eq. 1 cannot be written in the form presented in Eq. 2, as that is an approximation involving a learned critic. Further, it is odd to write the policy objective as directly maximising the learned Q-function–the policy-gradient maximises the observed reward via an advantage estimate A = Q' + r - Q.

More central to the proposal, I find the proposed attention mechanism rather different from what I expected. The authors cite the Attention is All You Need paper and use the same nomenclature, but rather than the usual softmax look-up operation, they use an MLP to compute the Query-Key look-up and match the values by an element-wise product. While that is valid, given that the attention mechanism is a well-established concept, I would expect such a departure from the norm to be well-motivated, but the authors only present it as is.

The imprecision of technical descriptions and lack of motivations become a serious problem when trying to parse the loss functions introduced in Eqs. 7-10. In Eq. 7, the authors minimise a loss function with respect to $\phi$, but it is not clear to me where these parameters are used. I understand that they define the parameters of the policy, but looking at Eq.5, are they also part of the $\bar{\pi}$, or does that function have separate parameters? In Eq.8, the authors mention training data, but don't define it. How does training data and validation data differ? Is it just two separate draws from the replay, or is there a separate validation replay that is being used? Similarly, after Eq. 9, the authors motivate their two-stage approach by "this allows the agent to obtain better actions by simply modifying the features compared to the original action without increasing the batch size, strengthening the agent's learning of good actions and reducing the probability of poor actions after the gradient step." I am still not sure what is meant by this, isn't it a policy-gradient step on the agent's policy parameters? For the attention loss in Eq. 10, could the authors please commend on why simply taking the policy-gradient of the post-adaptation actor-critic loss is not sufficient?

Overall, while I do appreciate the novelty of the idea and some attempts on the author's part to empirically demonstrate that their attention mechanism does provide non-trivial benefits, I feel that the authors have mainly presented a new architecture without proper motivation for the involved design choices, nor any empirical or theoretical results to motivate the components. Empirical results on the method as a whole only indicate moderate improvement that is within the statistical confidence interval of the baseline. As such, they are not sufficient to conclude that the proposed method represents a significant improvement for actor-critic methods.

Smaller nits:
- The algorithm chooses an action according to the highest Q-value if the 'actor dances twice'. I'm assuming this is a typo.
- I cannot parse the first sentence of paragraph 5.
- Final sentence, second paragraph on page 1 is missing a period.

**Summary Of The Paper:**

This paper proposes an attention-based actor-critic agent. The authors propose to parameterise the actor and the critic with two separate neural networks. Their algorithm operates in a two-stage fashion. In the first stage, the algorithm is a standard actor critic that produces an action probability distribution and a prediction of the action-value function. In the second state, the algorithm attends over the policy's and value function's hidden features. This output of the attention mechanism is a new feature representation that is fed into the policy's and value function's final layer, to produce an alternative action probability distribution and action value-function. The agent then takes the action that has the highest predicted return. The authors evaluate their proposed algorithm under DDPG and SAC on standard continuous control tasks and demonstrate that it can improve performance.

**Summary Of The Review:**

This paper proposes an attention-based architecture for actor-critic algorithms. While the idea is novel and has interesting elements to it, this submission is not ready for publication. The presentation is imprecise, to the point of obfuscating what is being optimised, while the motivation is either non-existent or makes sweeping claims that has no clear support. Empirically, the results do not sufficient to conclude that the proposed method presents an improvement for actor-critic algorithms.

---

### Official Review · Reviewer_qn99 · 2021-11-03

**Correctness:** 3
**Technical Novelty And Significance:** 2
**Empirical Novelty And Significance:** 2
**Recommendation:** 3
**Confidence:** 3

**Main Review:**

The main weakness of this work is its clarity. This work would severely benefit from adding high-level motivation / justification behind the design choices of the proposed architecture. For example, it's unclear what the goal of adding attention to these actor-critic architectures is -- is it just to improve the expressivity of the model? Or are explicit inductive biases that this work is trying to add in? The rationale behind the design choices for the attention are also unclear. Why is it reasonable to attend over the actor features using the Q-values as the key? Without laying the high-level ground work, it's difficult to understand what this work is trying to accomplish, and exactly what it's doing. It's worth noting that the paper does try to provide some high-level motivation and states:

> In the human decision-making process, people often modify their concepts based on the feedback and results to obtain a better decision. Inspired by this decision-making process, we use meta attention to adapt the features generated by the policy network based on the evaluation of value network.

However, it's still not entirely clear to me how this justifies the proposed attention mechanism (feedback and results seem more closely related to reward signal than a critic), and the concrete design decisions remain unjustified.

Additionally, in several areas, it would be helpful to be more precise. For example, line 4 of the algorithm 1 pseudocode states "if the actor dances twice," in reference to the top of page 6. It's somewhat difficult to understand precisely what the algorithm is doing here, although from the surrounding context, I gather that the algorithm just allows for sampling actions under two policy architectures and selecting the one with higher Q-value under the critic.

As a minor comment, the notation is also somewhat confusing. I would urge the authors not to overload $\pi$ with several variants, such as $\hat{\pi}$ as the last layer of the actor network and $\bar{\pi}$ for the feature representation, which makes it difficult to keep track of what means what.

The experimental results seem promising, although the improvement of the proposed approach over baselines is fairly modest in many tasks. Additionally, the plots are cutoff while many of the curves appear to still be learning, so it's difficult to gauge if the proposed approach indeed achieves a higher final performance. I would recommend running these curves for slightly longer until they've actually plateaued.

**Summary Of The Paper:**

This work proposes a new architecture that adds attention to actor-critic RL methods. This new architecture is evaluated on several Roboschool tasks.

**Summary Of The Review:**

I do not believe this work is ready for publication in its current form due to significant clarity issues. I was not able to fully understand all the details of the proposed approach, so it is possible that I've missed some crucial details, but I don't think its current presentation is sufficiently clear.

---

### Decision · Program_Chairs · 2022-01-20

**Decision:**

Reject

**Comment:**

The manuscript proposes a meta-attention based mechanism for improving off-policy actor-critic algorithms. Instead of introducing attention into networks at the level of pixels or multiple sources of information, this work focuses on using attention between features from the actor network (which become queries and values) and features from the critic network (which act as keys). Attention produces new features that are given as input to the action net, enabling it to potentially improve it's action selection. The attention is trained using a meta-learning objective that encourages outputting features that help other parts of the architecture to learn.
Reviewer note that there are some positive features of the paper. It is relatively well written and the figures are useful for spelling out the approach. In addition, so felt that the basic idea is an interesting one. However, there was general agreement that the manuscript is not ready for publication.
Most reviewers noted that, the newly proposed architecture and learning rules were not well motivated by the manuscript. Why was this particular approach pursued? Is there any better theoretical justification than can be offered? These questions on their own are not problematic. However, the empirical work does not robustly demonstrate that the algorithm yields a clear performance gain over baseline actor-critic methods in the literature. Most of the tasks are relatively simple and those gains that are observed are marginal. This is especially difficult for the manuscript given the increased compute and complexity of implementation required by the method. Finally, several reviewers were concerned about the presentation of some of the technical aspects of the works, potentially making it difficult to replicate important aspects of the work. In sum, the manuscript is not ready for publication.